

# Comparisons of genome assembly tools for characterization of *Mycobacterium tuberculosis* genomes using hybrid sequencing technologies

Kanwara Trisakul[1,2], Yothin Hinwan[1,2], Jukgarin Eisiri[2], Kanin Salao[1,2], Angkana Chaiprasert[3], Phalin Kamolwat[4], Sissades Tongsima[5], Susana Campino[6], Jody Phelan[6], Taane G. Clark[6,7] and Kiatichai Faksri[1,2]

[1] Department of Microbiology, Faculty of Medicine, Khon Kaen University, Khon Kaen, Thailand
[2] Research and Diagnostic Center for Emerging Infectious Diseases (RCEID), Khon Kaen University, Khon Kaen, Thailand
[3] Office for Research and Development, Faculty of Medicine Siriraj Hospital, Mahidol University, Bangkok, Thailand
[4] Division of Tuberculosis, Department of Disease Control, Ministry of Public Health, Bangkok, Thailand
[5] National Biobank of Thailand, National Center for Genetics Engineering and Biotechnology, Pathum Thani, Thailand
[6] Faculty of Infectious and Tropical Diseases, London School of Hygiene & Tropical Medicine, University of London, London, United Kingdom
[7] Faculty of Epidemiology and Population Health, London School of Hygiene & Tropical Medicine, University of London, London, United Kingdom

Corresponding author
Kiatichai Faksri, kiatichai@kku.ac.th

## ABSTRACT

**Background:** Next-generation sequencing of *Mycobacterium tuberculosis*, the infectious agent causing tuberculosis, is improving the understanding of genomic diversity of circulating lineages and strain-types, and informing knowledge of drug resistance mutations. An increasingly popular approach to characterizing *M. tuberculosis* genomes (size: 4.4 Mbp) and variants (*e.g.*, single nucleotide polymorphisms (SNPs)) involves the *de novo* assembly of sequence data.

**Methods:** We compared the performance of genome assembly tools (Unicycler, RagOut, and RagTag) on sequence data from nine drug resistant *M. tuberculosis* isolates (multi-drug (MDR) $n = 1$; pre-extensively-drug (pre-XDR) $n = 8$) generated using Illumina HiSeq, Oxford Nanopore Technology (ONT) PromethION, and PacBio platforms.

**Results:** Our investigation found that Unicycler-based assemblies had significantly higher genome completeness (~98.7%; $p$ values = 0.01) compared to other assembler tools (RagOut = 98.6%, and RagTag = 98.6%). The genome assembly sizes (bp) across isolates and sequencers based on RagOut was significantly longer ($p$ values < 0.001) (4,418,574 ± 8,824 bp) than Unicycler and RagTag assemblies (Unicycler = 4,377,642 ± 55,257 bp, and RagTag = 4,380,711 ± 51,164 bp). RagOut-based assemblies had the fewest contigs (~32) and the longest genome size (4,418,574 bp; *vs*. H37Rv reference size 4,411,532 bp) and therefore were chosen for downstream analysis. Pan-genome analysis of Illumina and PacBio hybrid assemblies revealed the greatest number of detected genes (4,639 genes; H37Rv reference contains 3,976 genes), while Illumina and ONT hybrid assemblies produced the highest number of SNPs. The number of

**PeerJ** ___________________________________________

genes from hybrid assemblies with ONT and PacBio long-reads (mean: 4,620 genes) was greater than short-read assembly alone (4,478 genes). All nine RagOut hybrid genome assemblies detected known mutations in genes associated with MDR-TB and pre-XDR-TB.

**Conclusions:** Unicycler software performed the best in terms of achieving contiguous genomes, whereas RagOut improved the quality of Unicycler's genome assemblies by providing a longer genome size. Overall, our approach has demonstrated that short-read and long-read hybrid assembly can provide a more complete genome assembly than short-read assembly alone by detecting pan-genomes and more genes, including IS*6110*, and SNPs.

# INTRODUCTION

Tuberculosis disease (TB), caused by the bacillus *Mycobacterium tuberculosis* (*Mtb*), is a major global health burden, with an estimated 10.6 million cases and 1.3 million deaths in 2022 alone (*World Health Organization, 2013b*). Drug-resistant (DR) *Mtb* makes the control of TB difficult. Multidrug-resistant TB (MDR-TB) refers to *Mtb* that is resistant to isoniazid (INH) and rifampicin (RIF). Pre-extensively drug-resistant TB (pre-XDR-TB) refers to a strain being MDR-TB and resistant to any fluoroquinolone (FQ). These serious forms of DR-TB can lead to poor patient outcomes (*World Health Organization, 2021*).

The generation and analysis of whole genome sequencing (WGS) data have become increasingly important in the study of bacterial population genetics. A recent development has been the wider application of *de novo* assembly methods to short and long sequence read data, as opposed to mapping to the reference genome H37Rv, for the characterization of *Mtb* isolate genomes and variants. Illumina short-read sequencing has revealed an extensive number of mutations associated with DR and compensatory effects. However, it can produce highly fragmented genome assemblies (*Chen, Erickson & Meng, 2020*), especially in unambiguously resolving long repeats present in multiple copies and GC-rich regions (H37Rv: GC content 65.6%). Despite the relatively clonal nature of the *Mtb* genome with no plasmids, analysis of Illumina short-read sequencing data commonly relies on alignment to the H37Rv reference genome. Short-read WGS analysis also has a limited ability to detect gene duplications, large structural variants, or variants in repetitive regions, such as in *pe/ppe* genes (*Bainomugisa et al., 2018*; *Gómez-González et al., 2023*). Differences in the software and parameters used in bioinformatic pipelines can affect the variants. Furthermore, input sequencing data ("reads") may come from various sequencing platforms, meaning that control strains with well-characterized variants among sequencing platforms are needed.

Third-generation sequencing technologies, such as Pacific Biosciences (PacBio) and Oxford Nanopore Technology (ONT) platforms, have the advantage over the Illumina short read sequencing by read length feature and can be used to facilitate the assembly of

complete *Mtb* genomes (*Gómez-González et al., 2023*; *Thorpe et al., 2024*). These sequencing platforms rely on single-molecule sequencing technologies, which have the ability to span repetitive regions in bacterial genomes, resulting in less fragmented or even complete genomes and thereby facilitating genome assembly independent from the reference strains. However, they can be more expensive (*e.g.*, PacBio) or have higher error rates of nucleotides (*e.g.*, ONT) compared to short-read sequencing technology. The long-read analysis can provide more comprehensive information on the evolutionary processes that led to the emergence of highly transmissible DR-TB strains (*Didelot et al., 2012*). However, studies of DR-TB strains with long-read sequencing technology are much less frequent than those involving short-reads, but the acceleration in the use of ONT platforms will mean robust pipelines for long-read analysis are urgently needed.

A hybrid assembly strategy using both long-and short-read data has been developed to improve genome characterization (*Chen, Erickson & Meng, 2020*). Long reads can scaffold contigs generated by Illumina short reads to correct assembly regions that cannot be resolved by Illumina short reads alone (*Chen, Erickson & Meng, 2020*). Unicycler (*Wick et al., 2017*) is a tool for assembling bacterial genomes from a combination of short and long reads that produces more accurate, complete, and cost-effective assemblies. The software constructs an initial assembly graph from short reads with the *de novo* assembler SPAdes, then simplifies the graph using information from both short and long reads. Reference-Assisted Genome Ordering UTility (RagOut) (*Kolmogorov et al., 2014*) and RagTag (*Alonge et al., 2022*) are recent software tools that can construct scaffolds and improve the quality of genome assemblies from Unicycler. Comparisons among these assembly tools are also limited, especially when applying hybrid assembly analysis among sequencing technologies.

In this study, we aimed to compare the performance of three genome assembly tools (Unicycler, RagOut, and RagTag) for genomic analysis of sequence data across *Mtb* isolates causing DR-TB. We developed a hybrid assembly genome approach across DR-TB strains for applications to a combination of Illumina short-read, ONT long-read, and PacBio long-read WGS data.

## MATERIALS AND METHODS

### Study population and *Mtb* sub-culture

We selected the DR-TB isolates based on the definition of pre-XDR-TB, where they are MDR-TB with additional resistance to any fluoroquinolone (FQ) (*World Health Organization, 2013a*). The isolates with concordant results between the proportional method (*Canetti et al., 1969*) and the minimum inhibitory concentration (MIC) test (using Sensititre MYCOTB MIC plates) were included. One MDR-TB and eight pre-XDR-TB isolates were selected. The phenotypic drug susceptibility test results are presented (Table S1), and were reported in our previous study (*Nonghanphithak et al., 2020*). Genotypic drug susceptibility testing and lineage classification were determined using TB-Profiler version 4.4.0 (*Coll et al., 2015*). Each isolate was inoculated on Löwenstein-Jensen (LJ) media and incubated at 37 °C for 4 to 8 weeks. A total of 2–3 loopfuls of colonies for each strain will be collected in a 16 × 150 mm tube with a few drops of sterile

nuclease-free water and sterile 6–8 of 5 mm glass beads and vortexed to break down the clumping colonies and bacterial cell wall. The tube was then be filled with 1 mL of sterile nuclease-free water and vortexed again. The bacterial cells were heat-killed at 80 °C for 30 min, cooled to room temperature, and 400 μL was aliquoted into two micro-centrifuge tubes for each isolate. The study protocol was approved by the Center for Ethics in Human Research, Khon Kaen University (HE661441).

## High-molecular-weight DNA extraction

High-molecular-weight DNA extraction was done using a modified method of the cetyltrimethyl-ammonium bromide-sodium chloride method (CTAB) (*de Almeida et al., 2013*). Fifty μL of 10 mg/mL lysozyme was added to each tube containing heat-killed bacterial cells and incubated at 37 °C overnight. Seventy μL of 10% SDS and 10 μL of 10 mg/mL proteinase K were added and incubated at 65 °C for 10 min. One hundred μL of 5 M NaCl and 100 μL of CTAB/NaCl, which were pre-warmed at 65 °C, were added, gently mixed until they became a milky solution, and incubated at 65 °C for 10 min. Then, 750 μL of chloroform/isoamyl alcohol (24:1) was added and gently mixed for at least 10 s, then centrifuged at 12,000 rpm (12,879 *g*) at 4 °C for 15 min. Then, 500 μL of supernatant was transferred into a new tube, and 10 μL of 10 mg/mL RNase A was added to each tube and incubated at 37 °C for 1 h. Then, 1 mL of cold absolute ethanol was added, the tube was inverted gently several times to mix, and then placed at −20 °C overnight. The DNA pellet was collected by centrifugation at 12,000 rpm (12,879 *g*) for 15 min at 4 °C, and the supernatant was discarded. The DNA pellet was rinsed three times with 1 mL of 70% ethanol and centrifuged at 12,000 rpm for 15 min at 4 °C. The ethanol was discarded, and the DNA pellet was left to dry at room temperature before being dissolved in 25 μL of nuclease-free water. DNA quality and concentration were measured using the Nanodrop 2000 and Qubit dsDNA HS assay kits (both from Thermo Fisher Scientific, Waltham, MA, USA).

## Whole genome sequencing

For short-read WGS, DNA samples were submitted to NovogeneAIT, Singapore, to generate 150-bp paired-end reads using the Illumina HiSeq sequencing platform. Long-read WGS were generated by OMICS DRIVE, Singapore, using the Oxford Nanopore Technologies (ONT) PromethION platform and SMRT sequencing from the Pacific Bioscience (PacBio) Sequel II sequencing platform. Whole genome sequencing data (.fastq files) have been deposited in the GenBank BioProject PRJNA598949, PRJNA598981, PRJNA613706, and PRJNA1021585.

## Genome hybrid assembly of nine DR-TB isolates using Unicycler, RagOut, and RagTag

Illumina short-read assembly (I), hybrid assembly of Illumina and Nanopore reads (IN), hybrid assembly of Illumina and PacBio reads (IP), and hybrid assembly of Illumina, Nanopore, and PacBio reads (INP) were first created using the default mode of Unicycler version 0.5.0 (*Wick et al., 2017*). For hybrid assemblies, SPAdes (*Bankevich et al., 2012*) was

used to create an Illumina short read assembly graph, and subsequently, Miniasm (*Li, 2016*) and Racon (*Vaser et al., 2017*) were used to construct bridges together with Nanopore-and/or PacBio long reads using the default parameters of Unicycler. All assemblies were submitted to perform genome-assisted *de novo* assembly using the default parameters of RagOut version 2.3 (*Kolmogorov et al., 2014*) and RagTag version 2.1.0 (*Alonge et al., 2022*) for scaffolding and improving genome assemblies. Assembly qualities (completeness, contamination, number of contigs, and genome size) were assessed using CheckM version 1.2.2 (*Parks et al., 2015*) and statistically compared pairwise using paired t-tests (*p* values < 0.05 were considered statistically significant). These quality analyses were compared using all available lineages in the *Mtb* taxonomy. The accuracy and identity with the reference genome H37Rv (NC_000962.3; size 4.4 Mbp, GC content 65.6%) of the assemblies were determined using nucmer and dnadiff (MUMmer version 4) (*Kurtz et al., 2004*). Assembly accuracy was statistically compared using one-way ANOVA and *post-hoc* tests (*p* values < 0.05 were considered statistically significant). Genome assembly graphs were visualized by Bandage software (*Wick et al., 2015*). A scheme of analysis workflow to generate the genome assembly of nine DR-TB isolates was presented (Fig. S1).

The complete genome assemblies (.fasta files) have been deposited in the GenBank BioProject PRJNA1021585, PRJNA1116704, and Supplemental Data.

## Pan-genome analysis

The RagOut I, IN, IP, and INP assemblies of nine DR-TB isolates were used for pan-genome analysis. Prokka (v1.14.0) software (*Seemann, 2014*) was used to annotate genome sequences before performing pan-genome analyses. The pan genomes were analyzed with Roary 3.12.0 (*Page et al., 2015*) using the genome annotations from Prokka as input to identify the number of core, soft-core, shell, and cloud genes in genomes. Venn diagrams comparing a number of genes identified from I, IN, IP, and INP assemblies were performed using an in-house Python script (https://github.com/jkeisiri/Matplotlib_Venn3-Gene).

## Mutation-associated drug resistance detection

Low-quality data were trimmed using Trimmomatic software (v0.38) (*Bolger, Lohse & Usadel, 2014*) for short reads and NanoFilt (*De Coster et al., 2018*) for long reads. Then, DNA sequences were mapped to the H37Rv reference genome (NC_000962.3) using BWA-MEM (v0.7.12) (*Li, 2013*). We used BCFtools (v1.9) (*Danecek et al., 2021*) to call variants, focusing on SNPs and insertions and deletions (indels). VCF files were used to generate the combined nucleotide frequencies among strains at each SNP position.

The mutations associated with resistance were detected from the combined nucleotide files using an in-house Python script (https://github.com/jkeisiri/Mtb_resist_gene_finding). Venn diagrams comparing the number of SNP detected from I, IN, IP, and INP assemblies were performed. An analysis workflow for pan-genome analysis and mutation-associated drug resistance detection was described (Fig. S2).
## RESULTS

### Drug-resistant *Mycobacterium tuberculosis* strains with phenotypically determined drug susceptibility profiles

The study considered one MDR-TB and eight pre-XDR-TB strains, whose drug susceptibility profiles were determined experimentally (including agar proportion and MIC test) (Table S1). Seven of the isolates are Beijing strains (sub-lineages 2.2.1), with signature large deletions identified (RD105, RD207, and RD181). The remaining two isolates were from lineages 2.1 (RD105) and 4.5 (RD122) (Table S1). Sequence data was generated for one MDR-TB and eight pre-XDR-TB strains across platforms (Illumina HiSeq, ONT PromethION, and PacBio sequel II). The average depth of sequencing coverage was 231-, 258-, and 147-fold for Illumina HiSeq, ONT PromethION, and PacBio Sequel II, respectively. The raw data were used to assemble reference genomes using three genome assembly software tools (Table S2), with the bioinformatic pipelines summarized (see Methods; Figs. S1 and S2).

### Comparison of genome assemblers in MDR-TB and pre-XDR-TB strains

The performance of three genome assemblers (Unicycler, RagOut, and RagTag) was compared. According to the CheckM quality assessment of nine DR-TB isolates assembled by three consensus tools, the completeness (%) of genome assembly analyzed using Unicycler (98.7% ± 0.4) was significantly higher (*p* values = 0.01) than RagOut (98.6% ± 0.5) and RagTag (98.6% ± 0.5). The Bandage assembly graph visualizer was used to assess the circular contigs for the nine Unicycler genomes (Fig. 1). Illumina short-read assemblies (I) show several interconnected nodes but no circularized contigs, whereas the hybrid assembly with ONT and PacBio long-read sequencing data provides more circularized contigs. The most circularized contigs were presented by the hybrid assembly of Illumina and ONT reads (IN; 7/9). Bandage graphs for RagOut and RagTag genome assemblies had a linear relationship and no evidence of a circularized contig. Taken together, genome assemblies using Unicycler software performed the best in terms of achieving contiguous genomes.

In addition to Unicycler, we also applied RagOut and RagTag assembly tools for scaffolding and improving the quality of genome assemblies. The number of contigs generated from the RagOut assembly tool (31.7) was significantly lower (*p* values < 0.001) than from the Unicycler (44.4) and RagTag (44.4) assembly approaches. The genome size (bp) of genome assembly by RagOut was significantly longer (*p* values < 0.001) (4,418,574 ± 8,824 bp) than Unicycler (4,377,642 ± 55,257 bp) and RagTag (4,380,711 ± 51,164 bp) assemblies (Table 1). Therefore, RagOut genome assemblies were chosen for the downstream analysis. The performance and accuracy of RagOut genome assemblies were assessed. The accuracy (%) of short-read and long-read hybrid assemblies (mean: IN; 99.2%, IP; 98.8%, and INP; 98.8%) was significantly higher (*p* values < 0.001) than short-read assemblies (96.6%) (Table S3).

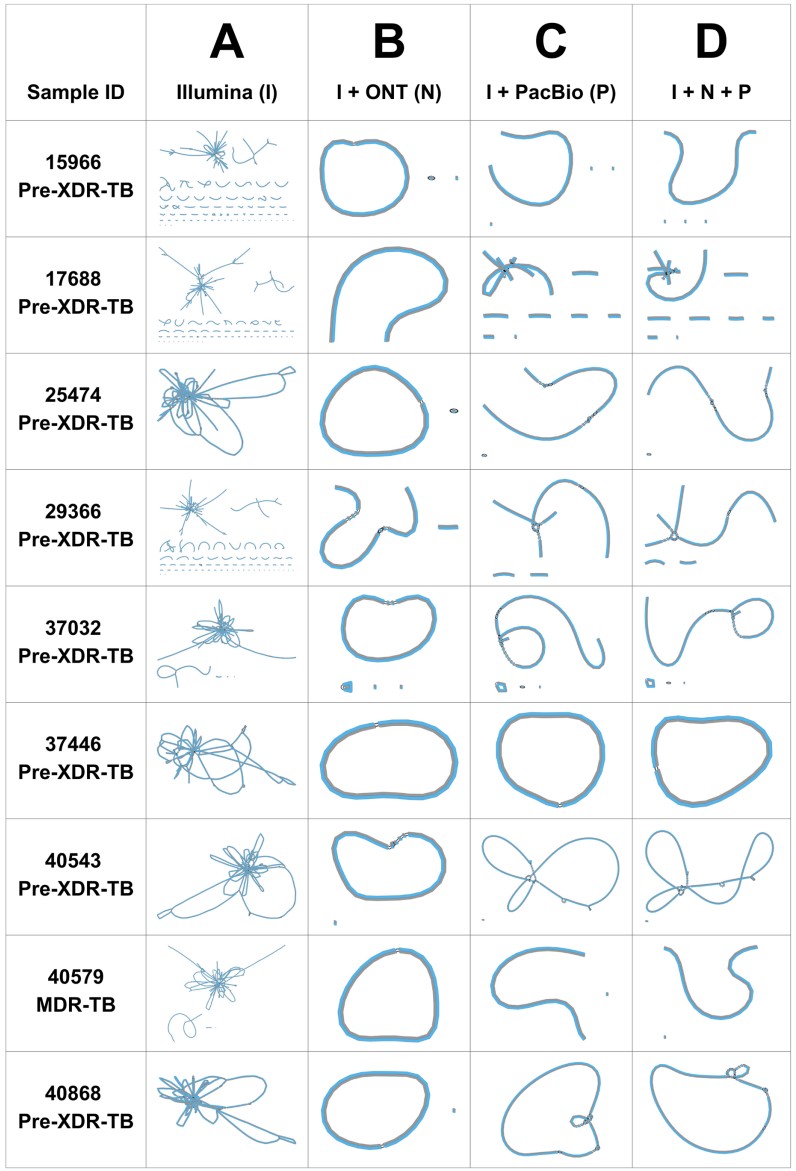

**Figure 1 Unicycler genome assembly graphs of nine DR-TB isolates, visualized by Bandage.** Unicycler genome assembly graphs of nine DR-TB isolates performed on (A) Illumina short-read assembly, (B) hybrid assembly of Illumina and Nanopore reads, (C) hybrid assembly of Illumina and PacBio reads, and (D) hybrid assembly of Illumina, ONT, and PacBio reads. Each line denotes a contig. The black connecting lines represent known overlaps.                

## Pan-genome analysis of MDR-TB and pre-XDR-TB strains using short-read and long-read assembly

From RagOut assemblies, the pan genomes of nine *Mtb* isolates of Illumina short-reads assembly (I) had 4,478 genes with 3,757 core genes and 721 accessory genes, hybrid assembly of Illumina and Nanopore reads (IN) showed 4,583 genes with 3,851 core genes and 732 accessory genes, hybrid assembly of Illumina and PacBio reads (IP) had 4,639 genes with 3,807 core genes and 832 accessory genes, and hybrid assembly of Illumina,

**Table 1 Quality assessment of genome assembly of nine DR-TB isolates obtained from the three consensus tools: Unicycler, RagOut, and RagTag.**

| Assembly | Completeness (%) | | | Contamination (%) | | | Number of contigs | | | Genome size (bp) | | |
|---|---|---|---|---|---|---|---|---|---|---|---|---|
| | Unicycler | RagOut | RagTag | Unicycler | RagOut | RagTag | Unicycler | RagOut | RagTag | Unicycler | RagOut | RagTag |
| I15966 | 98.8 | 98.4 | 98.4 | 0.3 | 0.3 | 0.3 | 191 | 153 | 191 | 4,240,573 | 4,416,758 | 4,256,173 |
| I17688 | 98.2 | 97.7 | 97.7 | 0.3 | 0.3 | 0.3 | 183 | 144 | 183 | 4,237,091 | 4,430,117 | 4,251,391 |
| I25474 | 98.9 | 98.7 | 98.7 | 0.3 | 0.3 | 0.3 | 124 | 81 | 124 | 4,344,155 | 4,424,835 | 4,352,555 |
| I29366 | 98.0 | 97.0 | 97.0 | 0.3 | 0.3 | 0.3 | 183 | 151 | 183 | 4,206,635 | 4,411,897 | 4,221,735 |
| I37032 | 98.9 | 98.8 | 98.8 | 0.3 | 0.3 | 0.3 | 130 | 82 | 130 | 4,346,887 | 4,428,600 | 4,355,087 |
| I37446 | 98.9 | 98.9 | 98.9 | 0.3 | 0.3 | 0.3 | 108 | 65 | 108 | 4,380,404 | 4,407,806 | 4,387,204 |
| I40543 | 98.9 | 98.8 | 98.8 | 0.3 | 0.3 | 0.3 | 136 | 78 | 136 | 4,354,500 | 4,430,659 | 4,362,700 |
| I40579 | 98.8 | 98.5 | 98.5 | 0.3 | 0.3 | 0.3 | 150 | 110 | 150 | 4,333,312 | 4,409,581 | 4,344,512 |
| I40868 | 98.9 | 98.8 | 98.8 | 0.4 | 0.3 | 0.3 | 113 | 67 | 113 | 4,353,990 | 4,424,057 | 4,360,990 |
| IN15966 | 98.9 | 98.9 | 98.9 | 0.3 | 0.3 | 0.3 | 3 | 1 | 3 | 4,405,217 | 4,401,767 | 4,405,217 |
| IN17688 | 98.8 | 98.8 | 98.8 | 0.3 | 0.3 | 0.3 | 1 | 3 | 1 | 4,402,178 | 4,402,557 | 4,402,178 |
| IN25474 | 98.9 | 98.9 | 98.9 | 0.3 | 0.3 | 0.3 | 2 | 1 | 2 | 4,426,760 | 4,414,055 | 4,426,760 |
| IN29366 | 97.7 | 97.7 | 97.7 | 0.6 | 0.6 | 0.6 | 7 | 7 | 7 | 4,361,631 | 4,422,807 | 4,362,131 |
| IN37032 | 98.9 | 98.4 | 98.9 | 0.3 | 0.3 | 0.3 | 6 | 6 | 6 | 4,426,144 | 4,422,517 | 4,426,444 |
| IN37446 | 98.9 | 98.9 | 98.9 | 0.3 | 0.3 | 0.3 | 1 | 1 | 1 | 4,431,145 | 4,431,145 | 4,431,145 |
| IN40543 | 98.9 | 98.9 | 98.9 | 0.3 | 0.3 | 0.3 | 7 | 8 | 7 | 4,419,996 | 4,420,062 | 4,420,496 |
| IN40579 | 98.8 | 98.8 | 98.8 | 0.3 | 0.3 | 0.3 | 1 | 1 | 1 | 4,419,214 | 4,419,214 | 4,419,214 |
| IN40868 | 98.9 | 98.9 | 98.9 | 0.6 | 0.6 | 0.6 | 2 | 1 | 2 | 4,421,737 | 4,418,471 | 4,421,737 |
| IP15966 | 98.8 | 98.8 | 98.8 | 0.3 | 0.3 | 0.3 | 4 | 2 | 4 | 4,396,818 | 4,409,638 | 4,396,818 |
| IP17688 | 98.2 | 98.1 | 98.1 | 0.3 | 0.3 | 0.3 | 18 | 16 | 18 | 4,340,985 | 4,417,797 | 4,342,385 |
| IP25474 | 98.9 | 98.9 | 98.9 | 0.3 | 0.3 | 0.3 | 10 | 9 | 10 | 4,404,879 | 4,425,619 | 4,405,579 |
| IP29366 | 98.0 | 98.0 | 98.0 | 0.3 | 0.3 | 0.3 | 18 | 14 | 18 | 4,344,880 | 4,418,077 | 4,346,180 |
| IP37032 | 98.9 | 98.9 | 98.9 | 0.3 | 0.3 | 0.3 | 14 | 12 | 14 | 4,410,922 | 4,429,026 | 4,411,622 |
| IP37446 | 98.9 | 98.9 | 98.9 | 0.3 | 0.3 | 0.3 | 1 | 1 | 1 | 4,430,612 | 4,430,612 | 4,430,612 |
| IP40543 | 98.9 | 98.9 | 98.9 | 0.3 | 0.3 | 0.3 | 38 | 24 | 38 | 4,405,319 | 4,415,430 | 4,407,419 |
| IP40579 | 98.6 | 98.6 | 98.6 | 0.3 | 0.3 | 0.3 | 2 | 2 | 2 | 4,408,919 | 4,416,618 | 4,408,919 |
| IP40868 | 98.9 | 98.9 | 98.9 | 0.4 | 0.3 | 0.4 | 20 | 11 | 20 | 4,398,429 | 4,403,054 | 4,399,429 |
| INP15966 | 98.8 | 98.8 | 98.8 | 0.3 | 0.3 | 0.3 | 4 | 2 | 4 | 4,396,818 | 4,409,638 | 4,396,818 |
| INP17688 | 98.2 | 98.1 | 98.1 | 0.3 | 0.3 | 0.3 | 18 | 16 | 18 | 4,340,985 | 4,417,797 | 4,342,385 |
| INP25474 | 98.9 | 98.9 | 98.9 | 0.3 | 0.3 | 0.3 | 10 | 9 | 10 | 4,404,879 | 4,425,619 | 4,405,579 |
| INP29366 | 98.0 | 98.0 | 98.0 | 0.3 | 0.3 | 0.3 | 18 | 14 | 18 | 4,344,880 | 4,418,077 | 4,346,180 |
| INP37032 | 98.9 | 98.9 | 98.9 | 0.3 | 0.3 | 0.3 | 14 | 12 | 14 | 4,410,922 | 4,429,026 | 4,411,622 |
| INP37446 | 98.9 | 98.9 | 98.9 | 0.3 | 0.3 | 0.3 | 1 | 1 | 1 | 4,430,612 | 4,430,612 | 4,430,612 |
| INP40543 | 98.9 | 98.9 | 98.9 | 0.3 | 0.3 | 0.3 | 38 | 24 | 38 | 4,405,319 | 4,415,430 | 4,407,419 |
| INP40579 | 98.6 | 98.6 | 98.6 | 0.3 | 0.3 | 0.3 | 2 | 2 | 2 | 4,408,919 | 4,416,618 | 4,408,919 |
| INP40868 | 98.9 | 98.9 | 98.9 | 0.4 | 0.3 | 0.4 | 20 | 11 | 20 | 4,398,429 | 4,403,054 | 4,399,429 |
| Mean | 98.7 | 98.6 | 98.6 | 0.3 | 0.3 | 0.3 | 44.4 | 31.7 | 44.4 | 4,377,642 | 4,418,574 | 4,380,711 |
| p-values | UvsRO | UvsRT | ROvsRT | UvsRO | UvsRT | ROvsRT | UvsRO | UvsRT | ROvsRT | UvsRO | UvsRT | ROvsRT |
| | 0.01* | 0.01* | 0.32 | 0.08 | 0.32 | 0.16 | <0.001* | N/A | <0.001* | <0.001* | <0.001* | <0.001* |

Notes:
* Statistically significant.
I, Illumina assembly; IN, Illumina+ONT assembly; IP, Illumina+PacBio assembly; INP, Illumina+ONT+PacBio assembly; U, Unicycler; RO, RagOut; RT, RagTag.

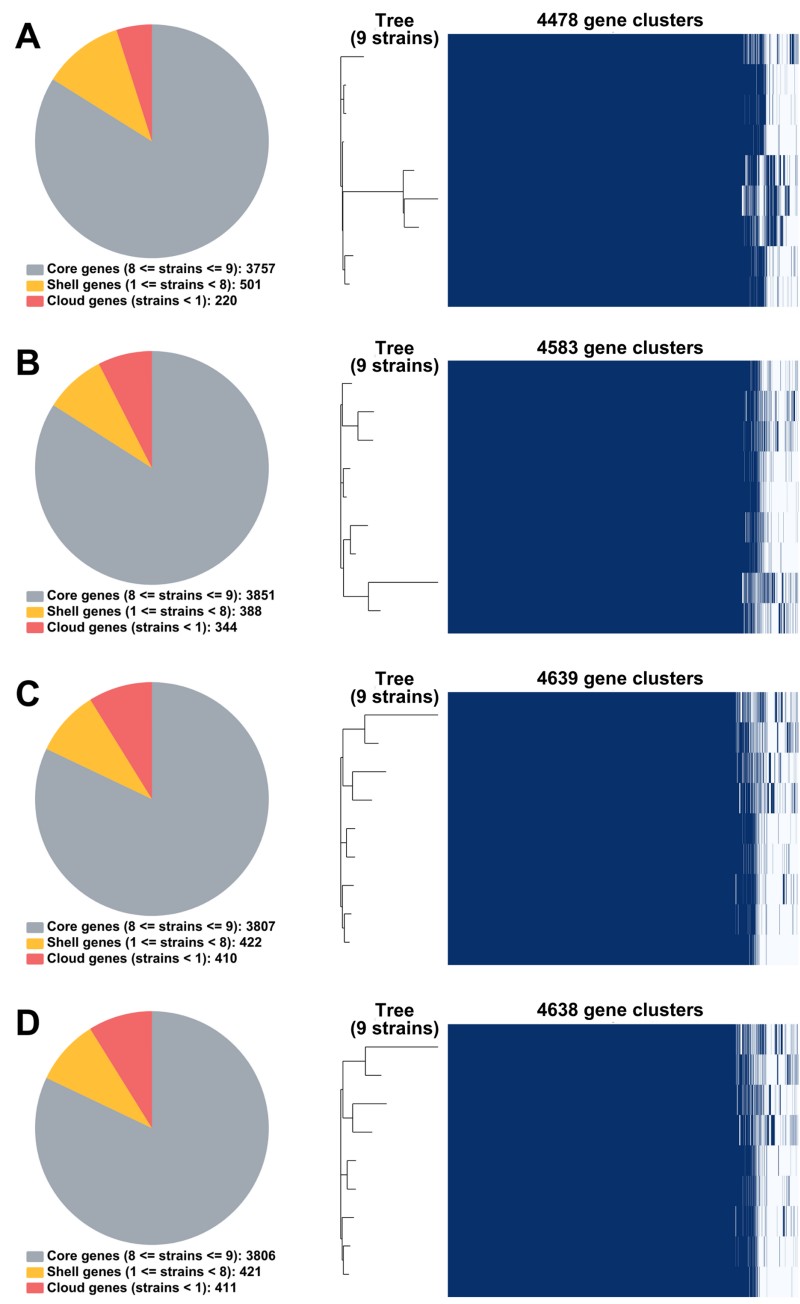

Figure 2 Pan genomes of nine DR-TB isolates by RagOut assemblies. (A) Illumina short reads, (B) hybrid of Illumina short reads and ONT long reads, (C) hybrid of Illumina short reads and PacBio long reads, and (D) hybrid of Illumina short reads, ONT long reads, and PacBio long reads.

ONT, and PacBio reads (INP) detected 4,638 genes with 3,806 core genes and 832 accessory genes (Fig. 2). Taken together, pan-genome analysis of IP assembly revealed the greatest number of detected genes.

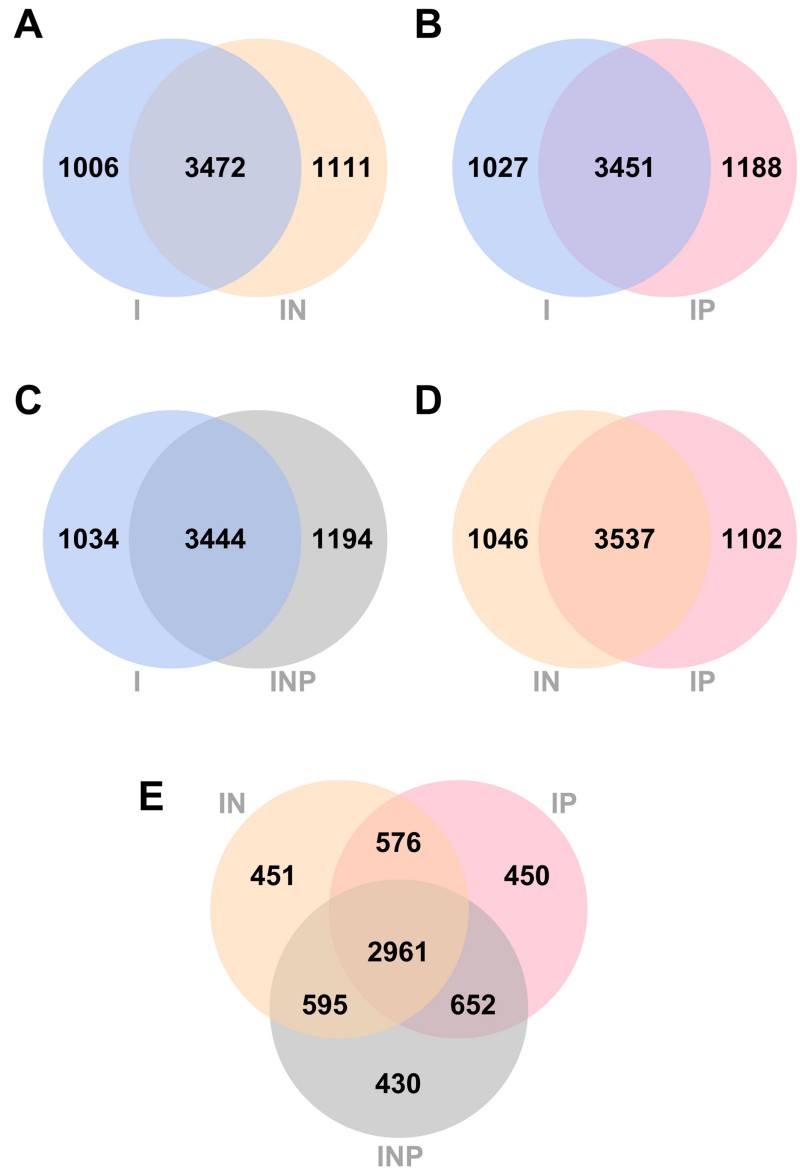

**Figure 3 Venn diagrams of the number of genes identified from RagOut assembly.** Venn diagram of (A) I and IN, (B) I and IP, (C) I and INP, (D) IN and IP, (E) IN, IP, and INP. The total number of identified genes in each platform assembly and the number of unique and common genes between them were shown. I; Illumina short reads assembly, IN; a hybrid of Illumina short reads and ONT long reads, IP; a hybrid of Illumina short reads and PacBio long reads, INP; a hybrid of Illumina short reads, ONT long reads, and PacBio long reads.

## Short-read and long-read hybrid assemblies provided more completed genome assembly than short-read assembly

The number of unique and common genes between I *vs.* IN, I *vs.* IP, I *vs.* INP, and among I, IN, and INP assemblies were compared. Venn diagrams of the number of genes detected across all Ragout assemblies are summarized (Fig. 3). The number of core genes and accessory genes detected in pan genomes of hybrid assemblies using ONT and PacBio long-reads (4,583 genes from IN, 4,639 genes from IP, and 4,638 genes from INP) was
higher than in short-read I assembly alone (4,478 genes). The unique gene sets identified from the Venn diagram belonged to hypothetical proteins and putative proteins, for example, in Figs. 3A, 3I (742/1,006 genes; 73.8%) and IN (759/1,111 genes; 68.3%) (Tables S4 and S5). The short-and long-read hybrid assemblies had a higher number of repetitive genes identified compared to the Illumina short-read assembly alone. For example, the insertion sequence (IS) family was found to be higher in I (8/1,006 genes; 0.8%) compared to IN (64/1,111 genes; 5.7%) (Tables S6 and S7). The copy number of IS*6110* in each strain was investigated. The short-and long-read hybrid assemblies detected the copy number of IS*6110* in most samples (6/9) from sub-lineage 2.2.1, whereas the Illumina short-read assembly alone did not detect any copy number of IS*6110* (Table S8).

### Detection of mutations in genes associated with MDR-TB and pre-XDR-TB from short-read and long-read hybrid sequence assemblies

The set of mutations has been selected from the TB-Profiler results of Illumina short-read WGS data of the nine DR-TB isolates, especially mutations in genes associated with resistance to any fluoroquinolones (FQs), in addition to isoniazid (INH) and rifampicin (RIF), according to the pre-XDR-TB definition. Genotypic resistant profiles and mutations in genes associated with MDR/pre-XDR-TB resistance in the nine DR-TB isolates are shown (Table 2). Thirteen resistance mutations were identified, including three mutations associated with INH resistance (*katG* Ser315Thr, Ser315Asn, and *fabG1* upstream T-8C), six mutations associated with RIF resistance (*rpoB* Leu452Pro, Ser450Leu, His445Leu, Asp435Val, His445Tyr, and *rpoC* Leu527Val), and four mutations associated with FQ resistance (*gyrA* Asp94His, Asp94Gly, Asp94Ala, and Ala90Val). Based on the I, IN, IP, and INP RagOut genome assemblies of nine DR-TB isolates, we investigated whether the hybrid assembly with ONT and PacBio long-read sequencing data could identify the mutation that covers all mutations in genes associated with MDR-TB and pre-XDR-TB. It was found that all assemblies could comprehensively identify (100%) mutations in candidate genes associated with MDR/pre-XDR-TB resistance in all isolates (Table 2). This result demonstrated the performance of hybrid assembly with ONT and PacBio long-read sequencing to identify all mutations in genes associated with MDR-TB and pre-XDR-TB.

Venn diagrams showing the number of total SNPs identified from all RagOut assemblies are presented (Fig. 4). Among nine DR-TB isolates, we found 4,942 combined variant positions from I assemblies, 10,455 combined variant positions from IN assemblies, and 10,109 combined variant positions from IP and INP assemblies. IN assemblies provide the greatest number of detected SNPs.

## DISCUSSION

WGS is becoming increasingly important in the study of *Mtb* genomics, especially in the serious forms of DR-TB, such as MDR-TB and beyond. Most of the available software and bioinformatics pipelines to predict anti-TB drug resistance using *Mtb* genomes derive mostly from Illumina short-read sequence data (*Coll et al., 2015*; *Phelan et al., 2016*; *Macedo et al., 2018*; *Hunt et al., 2019*). TB-Profiler can process long-read data (*e.g.*, ONT MinION (*Su et al., 2023*)). Furthermore, it is less common to analyse DR-TB within studies
**Table 2 Mutations in the candidate drug resistance gene used for identification of MDR-TB and pre-XDR-TB were identified from Illumina short-read sequencing data of DR-TB isolates by TB-Profiler.**

| Drug | Resistant genes | Locus tag | Mutation | Genomic position | Mutation in nine isolates | | | | | | | | | No. of strains found — Ragout assembly | | | |
|---|---|---|---|---|---|---|---|---|---|---|---|---|---|---|---|---|---|
| | | | | | 15966 Pre-XDR-TB | 17688 Pre-XDR-TB | 25474 Pre-XDR-TB | 29366 Pre-XDR-TB | 37032 Pre-XDR-TB | 37446 Pre-XDR-TB | 40543 Pre-XDR-TB | 40579 MDR-TB | 40868 Pre-XDR-TB | I | IN | IP | INP |
| INH | katG | Rv1908c | Ser315Thr | C2155168G | / | / | / | / | / | / | / | | / | 8/8 (100%) | 8/8 (100%) | 8/8 (100%) | 8/8 (100%) |
| | katG | Rv1908c | Ser315Asn | C2155168T | | | | | | | | / | | 1/1 (100%) | 1/1 (100%) | 1/1 (100%) | 1/1 (100%) |
| | fabG1 upstream | Rv1483 | T-8C | T1673432C | | | / | | | | | | | 1/1 (100%) | 1/1 (100%) | 1/1 (100%) | 1/1 (100%) |
| RIF | rpoB | Rv0667 | Leu452Pro | T761161C | / | | | | | | | | | 1/1 (100%) | 1/1 (100%) | 1/1 (100%) | 1/1 (100%) |
| | rpoB | Rv0667 | Ser450Leu | C761155T | | / | / | / | / | | | | | 4/4 (100%) | 4/4 (100%) | 4/4 (100%) | 4/4 (100%) |
| | rpoB | Rv0667 | His445Leu | A761140T | | | | | | / | | | | 1/1 (100%) | 1/1 (100%) | 1/1 (100%) | 1/1 (100%) |
| | rpoB | Rv0667 | Asp435Val | A761110T | | | | | | | / | | / | 2/2 (100%) | 2/2 (100%) | 2/2 (100%) | 2/2 (100%) |
| | rpoB | Rv0667 | His445Tyr | C761139T | | | | | | | | / | | 1/1 (100%) | 1/1 (100%) | 1/1 (100%) | 1/1 (100%) |
| | rpoC | Rv0668 | Leu527Val | T764948G | | | | / | | | | | | 1/1 (100%) | 1/1 (100%) | 1/1 (100%) | 1/1 (100%) |
| FQs | gyrA | Rv0006 | Asp94His | G7581C | / | | | | | | | | | 1/1 (100%) | 1/1 (100%) | 1/1 (100%) | 1/1 (100%) |
| | gyrA | Rv0006 | Asp94Gly | A7582G | | / | / | / | | / | | | | 4/4 (100%) | 4/4 (100%) | 4/4 (100%) | 4/4 (100%) |
| | gyrA | Rv0006 | Asp94Ala | A7582C | | | | | / | | | | | 1/1 (100%) | 1/1 (100%) | 1/1 (100%) | 1/1 (100%) |
| | gyrA | Rv0006 | Ala90Val | C7570T | | | | | | | / | | / | 2/2 (100%) | 2/2 (100%) | 2/2 (100%) | 2/2 (100%) |

**Note:**
I, Illumina assembly; IN, Illumina+Nanopore assembly; IP, Illumina+PacBio assembly; INP, Illumina+ONT+PacBio assembly; Isonoazid (INH); Rifampicin (RIF); Fluoroquinolones (FQs).
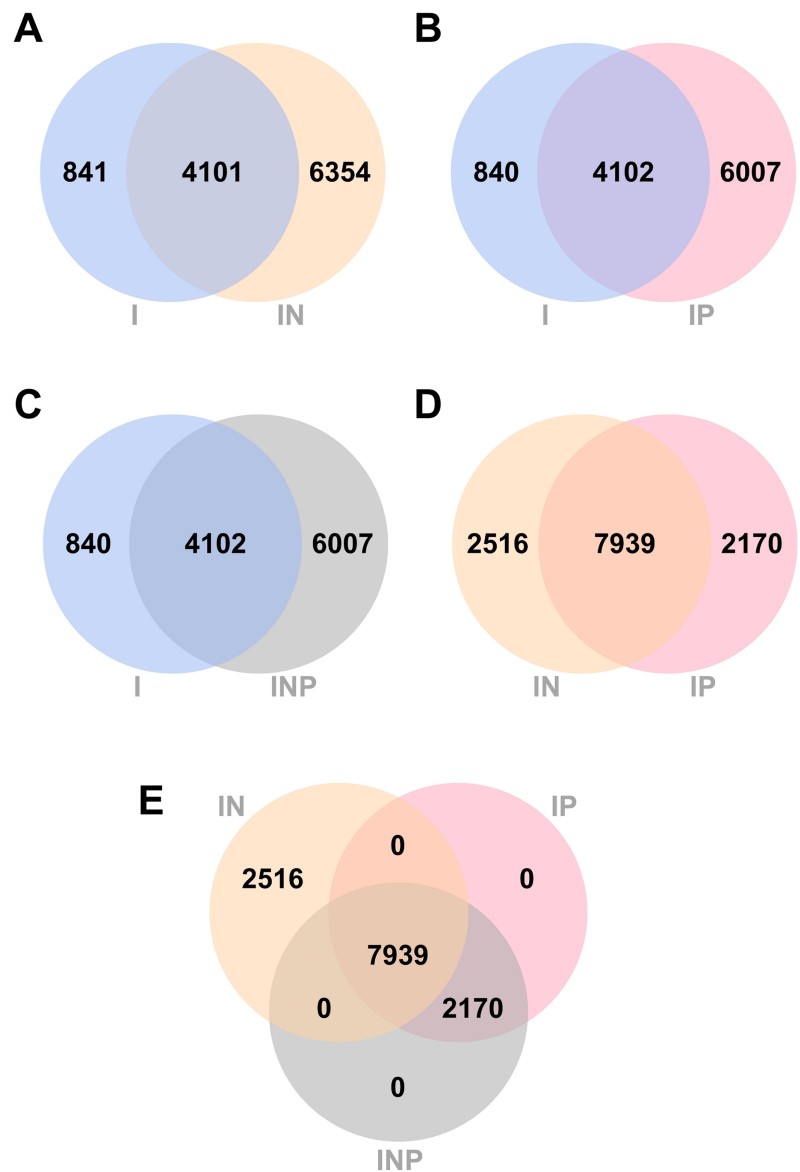

**Figure 4 Venn diagrams of the number of SNPs identified from RagOut assembly.** Venn diagram of (A) I and IN, (B) I and IP, (C) I and INP, (D) IN and IP, (E) IN, IP, and INP. The total number of identified SNPs in each platform assembly and the number of unique and common SNPs between them were shown. I; Illumina short reads assembly, IN; a hybrid of Illumina short reads and ONT long reads, IP; a hybrid of Illumina short reads and PacBio long reads, INP; a hybrid of Illumina short reads, ONT long reads, and PacBio long reads.               

based on a combination of Illumina, ONT, and PacBio WGS data (*Gómez-González et al., 2023*; *Thorpe et al., 2024*).

In this study, we analyzed the MDR-TB and pre-XDR-TB isolates using recent genomic assembly tools. The isolates used in this study are strains with well-characterized drug susceptibility test phenotypes based on the proportional method and MIC tests, which are completely in agreement with genotypic drug resistance profiles (WGS-based). Combined with long-read sequencing platforms, the assembly of complete genomes using a hybrid

assembly of Illumina short-read, ONT long-read, and PacBio long-read WGS data allows us to better characterize the genome structure of DR *Mtb*. This analysis approach allows us to compare the recently developed genome assembly software tools.

We generated the I, IN, IP, and INP hybrid genome assemblies of nine DR-TB isolates (one MDR-TB and eight pre-XDR-TB isolates) using three genome assembly tools (Unicycler, RagOut, and RagTag). Unicycler (*Wick et al., 2017*) is a tool for assembling bacterial genomes from a combination of short and long reads, resulting in more accurate, complete, and cost-effective assemblies. Unicycler uses SPAdes software (*Bankevich et al., 2012*) to create an initial assembly graph from short reads and then simplifies the graph using information from both short and long reads. From the quality assessment of all assembly results, we found that Unicycler had significantly higher genome assembly completeness (%) compared to other assembly tools, which could present as circular contigs in Bandage assembly graphs. While Illumina short-read assemblies show several interconnected nodes but no circularized contigs, ONT and PacBio long-read sequencing detected more circularized contigs when the hybrid assembly approach was employed. There are no circularized contigs presented by RagOut and RagTag genome assemblies, with all of them displayed in a straight-line graph. Genome assemblies constructed using Unicycler software performed the best in terms of achieving contiguous genomes. Our results are concordant with a previous study, which used Illumina and ONT sequencing to compare the hybrid assembly approaches of MaSuRCA, SPAdes, and Unicycler for ten bacterial strains (*Chen, Erickson & Meng, 2020*). They reported that Unicycler performed the best for achieving contiguous genomes, followed by MaSuRCA, while all SPAdes assemblies were incomplete. In our study, we further attempted to improve the quality of Unicycler's genome assemblies through reference-assisted *de novo* assembly with RagOut (*Kolmogorov et al., 2014*) and RagTag tools (*Alonge et al., 2022*).

We benchmarked the RagOut (*Kolmogorov et al., 2014*) and RagTag (*Alonge et al., 2022*) assembly tools for their performance in genome scaffolding and contrasted these with the quality of assemblies from Unicycler. We found that RagOut provided a significantly lower number of contigs and a longer genome size (bp) than RagTag. The lower number of contigs and longer genome size for Ragout may be due to algorithmic precision in the ordering of contigs, leading to their improved quality. A previous study demonstrated how to improve the quality of assemblies by connecting contigs into larger scaffolds and assisting assemblers in resolving ambiguities in repetitive regions of the genome using Ragout (*Kolmogorov et al., 2014*). In our study, RagOut assemblies from processed Unicycler assemblies of MDR-TB and pre-XDR-TB isolates provided the longest genome size (approx. 4,418,574 bp; compared to H37Rv, NC_000962, 4,411,532 bp). This finding supported the observation that RagOut provided the most complete genome assemblies across the three sequencing platforms. Therefore, we chose the RagOut genome assembler for downstream analysis to characterize the genome assembly of *Mtb*.

The pan-genome analysis could help us understand genetic diversity, the core genome, and the significance of many proteins encoded in the *Mtb* genome (*Dar et al., 2020*). We investigated the quality comparison of RagOut genome assemblies by pan-genome analysis to determine which assemblies from three sequencing platforms can provide the

greatest number of and most comprehensively detected genes. A previous study investigated the pan-genome analysis of ten bacterial species and reported a decrease in the number of core genes and an increase in the number of accessory genes in the pan-genome of low-compared to moderate-quality long-read assemblies (*Chen, Erickson & Meng, 2020*). In our pan-genome analysis, we demonstrated that those from RagOut assemblies of MDR-TB and pre-XDR-TB isolates had an increase in the number of core genes and accessory genes across all hybrid approaches compared to short reads alone. We found that the majority of the additional genes identified from hybrid short-and long-read assemblies belong to the insertion sequence (IS) family and tandem repetitive genes, especially IS*6110* and the *PE/PPE* family, compared to short-read assembly alone. IS*6110* is an insertion element specific to the *Mtb* complex species and frequently found inserted in a 36-bp array known as the Direct Repeat region (DR region: Rv2813-Rv2820c, RD207). IS*6110* is commonly found in multiple copies in East Asian lineages (lineage 2), especially Beijing strains (sub-lineage 2.2), and may not be found in some strains such as Indo-Oceanic (lineage 1) and Euro-American (lineage 4) (*Roychowdhury, Mandal & Bhattacharya, 2015*). However, the long-read sequencer is probably better at detecting these insertion elements. Our results showed that hybrid short-and long-read assemblies could detect the copy number of IS*6110* in the Beijing lineage host, especially sub-lineage 2.2.1 (6/9 samples). This demonstrated that adding long-read to short-read sequencing could help identify the increased number of genes coming from repetitive regions where long-read sequencing can identify them better (*Amarasinghe et al., 2020*).

Many studies have used a combined analysis of short and long reads. One study reported the novel variation in repetitive *pe/ppe* gene regions by assembling the complete genome using Illumina and Nanopore MinION sequencing data (*Bainomugisa et al., 2018*). Another study established two targeted-sequencing platforms for predicting DR in *Mtb* against 12 anti-TB drugs using Illumina MiSeq and Nanopore MinION. When compared to phenotypic drug susceptibility testing, both platforms achieved 94.8% sensitivity and 98.0% specificity (*Tafess et al., 2020*). Furthermore, online WGS analysis tools such as TB-Profiler (*Coll et al., 2015*) can be used to predict anti-TB drug resistance and *Mtb* lineage from WGS reads. Previous studies have benchmarked the capabilities of currently available software and bioinformatics pipelines for *Mtb* WGS data analysis and epidemiological links (*Jajou et al., 2019*), as well as their abilities to predict anti-TB drug resistance using a large data set of *Mtb* genomes derived primarily from Illumina short-read sequence data (*Coll et al., 2015*; *Phelan et al., 2016*; *Macedo et al., 2018*; *Hunt et al., 2019*). However, an analysis based on a combination of Illumina, ONT, and PacBio WGS data in DR-TB is still limited. Our results indicated that the improved performance of short-read combined with ONT and PacBio long-read assemblies could be attributed to superior hybrid assembly processes in which long-read sequences can compensate for the limitations of using only Illumina short-reads. The hybrid approach led to longer genome sizes (mean: ~55,216 bp longer) and more contiguous genomes, resulting in well-characterized MDR-TB and pre-XDR-TB pan genomes across three sequencing platforms. Moreover, we tested the complete genome of the hybrid assembly with ONT and PacBio long-read sequencing data to identify mutations in nine samples, including one

MDR-TB and eight pre-XDR-TB isolates. It was found that all the I, IN, IP, and INP RagOut genome assemblies could comprehensively identify (100%) mutations in candidate genes associated with MDR/pre-XDR-TB resistance in all isolates. Our well-characterized strains of the DR-TB could be used as resource sequences for DR prediction on short-read and long-read sequencing platforms.

A previous study compared the performance of three sequencing analysis approaches for investigating MDR-TB and pre-XDR-TB, including Illumina short-read assembly, ONT long-read assembly, and hybrid Illumina and ONT assembly (*Di Marco et al., 2023*). They reported that all three approaches agreed on identifying two major clusters, with hybrid assembly identifying more SNPs between the two clusters. When the quality of the assemblies was compared, hybrid and ONT long-read assemblies outperformed short-read assemblies. Here, our study further compared the performance of each hybrid assembly from Illumina short-reads combined with ONT and PacBio long-reads in downstream genomic analysis of MDR/pre-XDR-TB isolates. We investigated the I, IN, IP, and INP hybrid genome assemblies of one MDR-TB and eight pre-XDR-TB isolates. We found that the most circularized contigs were found by the IN hybrid assembly approach, the number of core genes and accessory genes detected in the pan genomes of hybrid assemblies using ONT and PacBio long-read was higher than in short-read I assembly alone, and the most combined variant positions were detected in IN assemblies (10,455 SNPs). These results indicated that short-read and long-read hybrid assembly provided more complete genome assemblies than short-read assembly alone. We also found that IP assembly pan-genome analysis revealed the greatest number of detected genes. A previous study suggested that, in terms of accuracy and completeness, hybrid assembly with either PacBio or ONT reads enabled high-quality genome reconstruction and outperformed long-read assembly alone (*De Maio et al., 2019*). Combining ONT and Illumina reads fully resolved most genomes at a lower consumable cost per isolate than PacBio. This represents a significant advancement in the hybrid genome assembly of Illumina short-reads and ONT long-reads, and it raises the question of whether PacBio long-reads are required for *Mtb* genomic studies due to sample preparation limitations and their high cost. However, the results of our pan-genome analysis showed that assembling complete genomes using a hybrid assembly of Illumina short-read, ONT long-read, and PacBio long-read WGS data can identify a greater number of detected genes and SNPs. Our results supported the utility of hybrid sequencing platforms and combined analysis in gaining a better understanding of genomic processes in *Mtb*.

Our study has some limitations. The ideal way to compare and standardize the samples among the three sequencing platforms is to use the same DNA sample input for all three sequencing platforms. However, in practice, it is difficult to use only a single DNA sample for all platforms, with Pacbio requiring excessive amounts. In the preparing DNA samples input step, nine DR-TB isolates were cultured and extracted using the manual CTAB method multiple times. Thus, obtaining enough high molecular weight (HMW) DNA from a single time of DNA preparation to meet long-read sequencing criteria is extremely difficult. We attempted to extract HMW DNA several times, sending each batch separately for sequencing with Illumina, Nanopore, and PacBio platforms. Consequently, the

variation in DNA sample input across the three sequencing platforms might affect the interpretation of our results. However, this confounding factor was not affected by either the comparison among the genome assembly tools or adding long-read sequencing to the short-read sequencing platforms. Another limitation is that the reference strain (H37Rv) was not sequenced across all platforms. While our comparative analysis using clinical samples supports our preliminary findings, incorporating H37Rv strains would undoubtedly provide a stronger control. Future studies sequencing H37Rv strains across these platforms could serve as an internal control for assessing algorithm performance.

Overall, our outlined approach is robust for the characterization of *Mtb* genomes and variants across a combination of sequencing platforms, and will inform the increasing use of *de novo* assembly methods involving long-read data to provide insights into drug resistance and transmission analysis.

## CONCLUSIONS

We compared genomic assembly tools (Unicycler, RagOut, and RagTag) using hybrid assemblies of nine DR-TB isolates (one MDR-TB and eight pre-XDR-TB isolates). Unicycler performed the best in terms of achieving complete genomes, while RagOut improved the quality of genome assemblies, providing a lower number of contigs and a longer genome size. We also demonstrated that hybrid assemblies among short-read and long-read sequencing technologies provided more completed genome assemblies with a better ability to detect IS*6110* and genes in the pan-genome than short-read assemblies alone. Such algorithmic insights will inform the robust application of WGS approaches for the control of TB.

### Funding

This work was supported by the National Research Council of Thailand (NRCT): NRCT5-RGJ63003-047 and NRC MHESI 483/2563 and the Research and Diagnostic Center for Emerging Infectious Diseases (RCEID), Khon Kaen University, Khon Kaen, Thailand. Taane G. Clark and Susana Campino are funded by UKRI MRC (MRC IAA2129, MR/R026297/1, and MR/X005895/1) and EPSRC (EP/Y018842/1) grants. The funders had no role in study design, data collection and analysis, decision to publish, or preparation of the manuscript.

### Grant Disclosures

The following grant information was disclosed by the authors:
National Research Council of Thailand (NRCT): NRCT5-RGJ63003-047 and NRC MHESI 483/2563.
Research and Diagnostic Center for Emerging Infectious Diseases (RCEID), Khon Kaen University, Khon Kaen, Thailand.
UKRI MRC: MRC IAA2129, MR/R026297/1, and MR/X005895/1.
EPSRC: EP/Y018842/1.

 

## Competing Interests

The authors declare that they have no competing interests.

## Author Contributions

- Kanwara Trisakul conceived and designed the experiments, performed the experiments, analyzed the data, prepared figures and/or tables, authored or reviewed drafts of the article, and approved the final draft.
- Yothin Hinwan analyzed the data, prepared figures and/or tables, and approved the final draft.
- Jukgarin Eisiri analyzed the data, prepared figures and/or tables, and approved the final draft.
- Kanin Salao conceived and designed the experiments, authored or reviewed drafts of the article, and approved the final draft.
- Angkana Chaiprasert conceived and designed the experiments, authored or reviewed drafts of the article, and approved the final draft.
- Phalin Kamolwat conceived and designed the experiments, authored or reviewed drafts of the article, and approved the final draft.
- Sissades Tongsima conceived and designed the experiments, authored or reviewed drafts of the article, and approved the final draft.
- Susana Campino analyzed the data, authored or reviewed drafts of the article, and approved the final draft.
- Jody Phelan analyzed the data, authored or reviewed drafts of the article, and approved the final draft.
- Taane G. Clark analyzed the data, authored or reviewed drafts of the article, and approved the final draft.
- Kiatichai Faksri conceived and designed the experiments, performed the experiments, analyzed the data, prepared figures and/or tables, authored or reviewed drafts of the article, and approved the final draft.

## DNA Deposition

The following information was supplied regarding the deposition of DNA sequences:

Whole genome sequencing data (.fastq files) are available at GenBank: PRJNA598949, PRJNA598981, PRJNA613706, and PRJNA1021585.

The complete genome assemblies (.fasta files) are available at GenBank: PRJNA1021585 and PRJNA1116704.

## Data Availability

Supplemental data is available in Supplemental Tables and Supplemental Figures.

The raw data of one MDR-TB and eight pre-XDR-TB genome assemblies in this study are shown in the Supplemental Tables.

The bioinformatic pipelines are summarized in Figs. S1 and S2.

## Supplemental Information

Supplemental information for this article can be found online at http://dx.doi.org/10.7717/peerj.17964#supplemental-information.

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
