# Peer review of "Comparisons of genome assembly tools for characterization of Mycobacterium tuberculosis genomes using hybrid sequencing technologies"

_PeerJ, doi:10.7717/peerj.17964_

## Round 0.1 · original submission · Minor Revisions

Dear authors, thank you for your submission. After minor revisions your manuscript will be ready for publication in PeerJ. Please, refer to the reviewers' comments for further details. I also urge you to proofread everything, also ensuring for enough quality/resolution in any figures as per guidelines (in case you have not provide so yet), before resubmission.

Reviewer 1 ·

Basic reporting

The authors compared popular hybrid genome assembly tool (i.e., Unicycler, RagOut, RagTag) on 9 drug resistant M. tuberculosis isolates, showing that Unicycler software performed the best in terms of contiguous genomes, and RagOut improved the quality of Unicycler's genome assemblies by providing a longer genome size.

Experimental design

1. In line 148, what is the sequencing depth and coverage for Illumina HiSeq platform?
2. In line 159, please mention the Unicycler parameters for SPAdes/Racon/Miniasm. Whether you use `--isolate` mode? How about the effects of different parameters on the results?
3. In line 161, default parameter for RagOut and RagTag?
4. In line 211-212: what statistical testing technique is used for the p value? Please introduce one section to the method part.

Validity of the findings

The findings are generally valid but please provide the statistical testing technique and software parameters.

Reviewer 2 ·

Basic reporting

L40:
Should read:
based on RagOut was significantly longer (p values < 0.001)
Placing the p-value after the word "longer" makes the sentence clearer and makes the context of the significance clear before seeing the specific p-value and assembly sizes.

L121:
This should be past tense and should read:
The tube was then be filled with 1 mL of sterile nuclease-free water and vortexed

L122-123:
This should be past tense and can be more concise, please consider rephrasing to:
The bacterial cells were heat-killed at 80°C for 30 minutes, cooled to room temperature, and 400 µL was aliquoted into 2 micro-centrifuge tubes for each isolate.

L129-131:
Inconsistent start of sentence with number spelt out and use of numbers. Please consider this revision:
“Fifty µL of 10 mg/mL lysozyme was added to each tube containing heat-killed bacterial cells and incubated at 37°C overnight. Seventy µL of 10% SDS and ten µL of 10 mg/mL proteinase K were added and incubated at...”

L138:
Be specific about the number of time “flipping”. Please consider revising:
"was added, the tube was inverted gently several times to mix, and then placed at -20°C overnight."

L158-159:
Consider rewording:
hybrid assemblies, SPAdes (15) was used to create an Illumina short read assembly graph, and subsequently, Miniasm (16) and Racon (17) were used to construct bridges

L212:
Please consider using ‘higher’ before stating the p-value:
“was significantly higher (p values = 0.01)”

L223-224:
Please consider using ‘lower before stating the p-value:
“was significantly lower (p values < 0.001) than”

L413-416:
Please consider revising to be more concise:
"We attempted to extract HMW DNA several times, sending each batch separately for sequencing with Illumina, Nanopore, and PacBio platforms. Consequently, the variation in DNA sample input across the three sequencing platforms might affect the interpretation of our results."

Experimental design

The experimental design is robust and well-structured, effectively comparing the performance of different genome assembly tools across various sequencing platforms.

Validity of the findings

The findings are valid and well-supported by comprehensive data analysis, demonstrating the advantages of hybrid assembly approaches for genome completeness and accuracy.

Additional comments

The study by Trisakula et al. demonstrates that hybrid assembly approaches can produce more complete and accurate genome assemblies of Mycobacterium tuberculosis. The improved genomic resolution offered by hybrid approaches have the potential to enhance our understanding of genomic diversity and drug resistance mutations, which is crucial for developing better diagnostic tools, treatments, and public health strategies to combat tuberculosis. The manuscript is clearly written, with well-defined methods and research questions, and well-stated conclusions.

Reviewer 3 ·

Basic reporting

No comment - The article is well written, concise and easy to follow.

Experimental design

One suggested improvement/inclusion of an analysis:
Why was the reference strain (H37Rv) not considered for analysis using the different approaches? This could have served as an elegant ''internal" control and a way to cross-validate these approaches.
Since epidemiology was not the goal of this manuscript, the inclusion of non-clinical sequences could have proved as a valuable addition.

Validity of the findings

No comment

Additional comments

Some minor edits to some figures are needed.
Figure 1:
Please amend Illumina 3 and 7-8, to ensure that the figure fits better into the column.
Figure 2: No soft-core genes are visible in the pie chart so it should be removed from the legends.

Other general comments:
Which version of TB-Profiler was used?

---

## Round 0.2 · accepted · Accept

Dear authors, congratulations. I am now accepting your work for publication.